# Early Interventions and Impact of COVID-19 in Spain

**DOI:** 10.3390/ijerph18084026

**Published:** 2021-04-12

**Authors:** Uxue Alfonso Viguria, Núria Casamitjana

**Affiliations:** 1Institute of Global Health, Université de Genéve, 1211 Geneva, Switzerland; 2ISGlobal, Hospital Clínic—Universitat de Barcelona, 08036 Barcelona, Spain; nuria.casamitjana@isglobal.org

**Keywords:** COVID-19, Spain, health systems, coronavirus, SARS-CoV2, case study, epidemiology, global health

## Abstract

The health crisis emerging from China in January 2020 has spread around the world resulting in a disruption of daily life activity in many countries. In response to this health threat, different measures have been implemented by national governments to minimize the possible health consequences. This article explores the initial impact of the COVID-19 pandemic in Spain, providing an epidemiological overview and reviewing the early measures developed to control its spread. On 1 April, Spain was the country with the second highest incidence in the world; with 104,118 positive cases detected and 9387 deaths recorded. Among these, 20.2% of positive cases were among healthcare professionals. In addition to the unprecedented health crisis, the lockdown interventions employed were considered to be among the strictest measures implemented through European countries. These measures were initially successful in controlling local transmission, but resulted in severe economic and social impacts. A critical review of the actions taken and their impact on the Spanish population could contribute to guide and inform decision-making in future pandemic situations.

## 1. Introduction

In December 2019, a new virus later known as severe acute respiratory syndrome coronavirus 2 (SARS-CoV-2) emerged in the city of Wuhan (China). Though the characteristics of the disease were initially uncertain, its transmission mechanisms, including long incubation periods and high proportion of mild and asymptomatic cases resulted in a rapid spread of the virus. This rapid spread resulted in significant difficulties for international as well as national and local health authorities to control the outbreak. On 30 January, the World Health Organization (WHO) declared the outbreak a Public Health Emergency of International Concern (PHEIC) after 82 cases of COVID-19 had been detected in 18 countries outside China [1]. On March 11th, after 118,000 cases and 4291 deaths were reported in 114 countries around the world it was declared a pandemic [2]. In Spain, the first case was detected the 31st of January on the island of La Gomera in the Canary Islands, and during the first weeks of February few positive cases were detected, initially only among tourists travelling from high risk areas (China, Iran, South Korea, Singapore, and Italy). However, during the month of February there was a significant increase in daily new cases and by 3rd of March, sustained transmission was confirmed in some areas of Spain [3]. The number of new cases increased exponentially through March, with the peak of the initial first wave of the outbreak being observed during the last week of March. On 30 April, after two months of sustained transmission, there were 213,435 positive identified cases, and 24,543 COVID-19-related deaths [4].

Across the world countries have provided a broad range of responses to the COVID-19 pandemic, taking into consideration their own population demographics, cultural characteristics and the different impact of the disease. As a result, we have been able to witness diverse policy strategies and responses at the epidemiological level. Some countries have been able to control the spread faster and reduce the number of positive cases detected daily in less than a month; while others have focused their policies on reducing the possible economic and social impact by maintaining less severe social distance measures. However, common in all affected countries is the fact that the impacts of the health crisis go beyond the severe health consequences and have strong effects on the economic and social wellbeing of their populations [5,6,7,8,9,10,11,12,13,14,15,16,17].

From the notification of the outbreak by China, national experts monitored its evolution and implemented multiple control measures in Spain. Although there has been some debate on the effectiveness and timeliness of these measures, it is clear that these interventions have resulted in significant impacts on the population that go beyond those that can be represented through epidemiological curves and rates of incidence and death. To better understand all the effects that will result from this health crisis, it is necessary to take a multidisciplinary perspective and consider all potential direct and indirect consequences for the population. This article aims to describe the interventions implemented in Spain in response to the COVID-19 outbreak, and its multiple impacts on the population.

## 2. Case Presentation

It is important to consider the characteristics of a country in order to understand how the COVID-19 outbreak and the interventions developed by the authorities affect the country and its citizens. Spain is a European country with 46.7 million inhabitants, and a population density of 94 people per km^2^. The life expectancy at birth is 83.4 years, one of the highest in Europe [18]. 60% of the population is 65 years and older and have at least one chronic disease, which in the context of the current epidemic are considered high risk groups for developing severe disease. According to OECD, per capita GDP is 41,724 USD [19]. The most predominant sectors in the Spanish economy are the services sector (75.8%), followed by the industrial sector (13.9%), construction (5.8%) and agriculture (4.4%). The unemployment rate has been progressively decreasing since the economic recession of 2008 and currently (April 2020) stands at 14.1% [20]. Spain is located in the southwestern region of Europe. The climatic characteristics are heterogeneous and variate through the periphery and interior of the peninsula. Temperatures in the interior present cold winters ranging from 0 to 3 °C, and warm summers with an average of 24 °C in July and August. In the Mediterranean littoral winter is mild with an average of 10 °C in January, and mean temperatures of 16 to 18 °C through the year. Spain has a total area of 505,983 km^2^, divided into 19 regional territories and 17 autonomous communities. The country is governed through a parliamentary constitutional monarchy by the king, Felipe IV, and a president, Pedro Sanchez Perez-Castejón. Spain is a decentralized unitary state with asymmetrical devolution of powers to each autonomy, which are governed according to the constitution and statutes of each autonomic region [21].

Management of healthcare services is decentralized through each autonomous community and harmonized by the Inter-territorial Council of the Spanish National Health Service. Health services are provided to all Spanish citizens and foreign residents in Spain free of charge. They are administered through the National Health System, defined by law as the coordinated set of health services delivered by each autonomous community. According to the “Multidimensional Framework for Rating Health System Performance and Sustainability”–which uses a range of ‘AAA’ to ‘CCC’ to classify health systems through an analysis of outcomes in three domains: general performance, clinical outcome, and sustainability-, the Spanish health systems is rated as ABA [22]. Health spending per capita reached 2371 euros in 2017, 15% below the EU average. Healthcare spending in Spain has decreased since 2008 due to the economic crisis, and since then there are claims that the health system is underfunded. Mortality rates for preventable and treatable causes are among the lowest compared to the EU average, indicating effective health interventions in avoiding premature mortality [23]. Nevertheless, the surge in healthcare demand caused by the COVID-19 pandemic has demonstrated weaknesses in the healthcare system. At the end of March 2020, Spain was one of the countries most significantly impacted by the COVID-19 outbreak. According to data retrieved by the Spanish Ministry of Health, from 24 February to 31 March of 2020, Spain recorded 102,136 positive cases with 9053 deaths. The average daily new cases recorded were 7400, and 800 daily deaths occurred during that period due to COVID-19 [24]. Health system weaknesses are most evident in the reduced hospital bed capacity, lack of supplies, and limited staffing. The Spanish health system currently has 3.0 hospital beds per 1000 population, lower than the EU average of 4.8. More critical in this health crisis has been the number of acute care beds available which in Spain is 9.7 per 100,000 population, lower than the OECD average of 12 and far lower than the German average of 33.9 beds per 100,000. The number of doctors in Spain is 3.9 per 1.000 population, slightly higher than the EU average (3.6 per 1.000); the proportion of nurses 5.7 per 1.000 population is below the average of 8.5 per 1000 [25]. Although hospitals were quickly transformed to increase their capacity and healthcare professionals worked tirelessly for several weeks, there was insufficient health system capacity to meet the increasing demands, which were observed during the initial peak of the pandemic.

## 3. Management and Outcome

Since the start of the outbreak the WHO has declared the event a Public Health Emergency of International Concern (PHEIC), following the International Health Regulations. According to the information released by the Health Ministry, a monitoring and surveillance system was implemented with the purpose of promptly detecting and controlling the spread of the virus into Spain. In addition, the capacities required to act in the event of a PHEIC were reviewed, resulting in the development of a protocol for action that was elaborated before the first positive cases were detected in Spain and constantly reviewed as the outbreak evolved [26]. However, these first protocols and plans of action were not publicly disclosed and there are some who still question whether the international alarms warning about the possible broad scale of the pandemic were duly considered by Spanish public health authorities [27]. According to the official press releases regarding the unpublished protocols, these set four main priorities: epidemiological surveillance, early detection, prevention of transmission, and contact tracing [28]. Nevertheless the first comprehensive plan of action is dated on the 16th of July [29]. After the outbreak spread into Europe, and first few cases were detected in Spain a second extraordinary inter-territorial council of the National Health Systems was called to coordinate the response at the national and territorial levels. This council took place on the 25th of February 2020. Following this council, initial measures were implemented to control the spread of the virus into Spain from neighboring countries. These included reinforced detection capacity and increased information distribution among the population, especially travelers [30]. At the same time, an inter-ministerial coordination committee was created to unify a cross-disciplinary response through ministries involved in the health emergency [31].

From the beginning of March, positive cases detected in Spain started increasing exponentially, and sustained community transmission was observed in many areas around the country. On March 9th, the first measures with broad impact for the population were implemented in areas with significant community transmission. At this point, only three regions were considered to have sustained community transmission: Madrid, the city of Vitoria, and the region of Labastida. These measures included the promotion of social distancing in the educational, working and social environments; closing schools and universities, promoting remote work, and limiting all events to a maximum of one thousand participants. Measures recommended for the population in all Spanish territories included encouraging homecare for the elderly and other vulnerable sectors of the population, avoiding crowded spaces, maintaining social distance of at least one meter, avoiding unnecessary travel, and staying home if symptoms developed [32]. These recommendations were reinforced with legal modifications to allow centralized purchase of medical equipment by the Health Ministry. Foreseeing the potential economic impact that the confinement measures could entail in the Spanish economy, the government established some temporary measures to protect small and medium businesses; in order to mitigate the negative effect of the health crisis on the population relying on small and medium enterprises. Financial assistance measures included flexibility in tax payments and specific financial aid for businesses in affected sectors [33].

On March 12th, confinement measures where extended to all the Spanish territories, and two days later a State of Alarm was declared for an initial period of fifteen days. This was later increased by an additional fifteen days until the 21st of June. The decree, which declared the State of Alarm, limited the freedom of movement of all the population in Spain; moreover, this decree forced suspension of all non-necessary commercial activity as well as recreational activities, tourism and food service industry [34]. Border control measures were also established, including a complete closure of admissions from Italy. In addition, on 21 March a Scientific and Technical Committee for COVID-19 was established by the president of the government and the health minister to provide advice on the interventions and measures to face the SARS-CoV2 outbreak. This is committee included six experts from different fields and led by Fernando Simón, director of the Center for Coordination of Sanitary Alerts and Emergencies (CCAES in Spanish) [35].

The government found difficult to implement the economic and social measures due to the internal political disagreement on the governing coalition. The Spanish political context is characterized by a strong division between socialists and populists, and the current government is administered by a coalition of two political parties. This conflicted political environment caused a delay in the implementation of the measures to control the disease. And resulted in some territorial administrations taking preliminary measures at the subnational level in an uncoordinated manner [36]. The lack of political agreement and coordination at the national and subnational level has continued through the different stages of the pandemic.

During this health emergency, each autonomous community was responsible for the management of health services in each territory. Complemented with centralized logistics from national authorities to supply material resources related to combating the COVID-19 outbreak, strategically planned to face the global shortage of protective equipment, testing devices and other equipment needed to treat and manage the disease- such as ventilators. Nevertheless, this centralized planning was criticized due to a perceived lack of experience in purchasing healthcare material by the Health Ministry and due to the lack of coordination with the territorial administrations [37].

In the early stages of the outbreak the lack of human and material resources on the healthcare system was of particular concern. To support the National Health System in its provision of health services to all the population, the government decided to increase the capacity of the National Health System with 52,000 healthcare professionals, including final year students and retired professionals; and the availability of the healthcare professionals and infrastructure from the private sector [38]. The General Council of Official Medical Professional Colleges (CGCOM) responded to these measures through an official statement claiming its opposition [39]. Through this official release, this collective of medical doctors expressed their worries of having professionals working before completing their medical training. In addition, they claimed the lack of participation of healthcare professionals in the development of the public health measures concerning the institutions in which they work. This statement evidenced the ineffective coordination between national and subnational authorities.

Together with the lack of human resources, lack of material supplies and infrastructure, particularly hospital capacity to treat patients, was of particular concern. Given the long periods of time under acute care needed by patients suffering severe Covid-19 illness, maximum bed and ICU capacity was exceeded in many territories. Territorial administrations responded by enabling temporary hospitals in hotel buildings and other public and private establishments. In response to the lack of sufficient medical equipment to meet the increasing demands of the healthcare sector, a wave of innovation and solidarity was triggered among multiple businesses, including those not specialized in medical equipment. An example of this corporative solidarity was the auto manufacturer SEAT, who contributed to the crisis by developing new prototypes of ventilators to counteract the shortage of these devices [40]. The global shortage of protective equipment notably affected healthcare professionals in Spain. Due to unavailability of personal protective equipment and facial masks, hospital workers were overexposed to the virus and this was reflected in the increased number of positive cases among healthcare workers. As of 20 April 2020, the percentage of infected healthcare professionals was estimated to be 20.2% of the total confirmed cases according to the report published by Instituto Nacional Carlos III and Centro Nacional de Epidemiología (ISC-CNE). Which later increased to 24.1% of infected healthcare professionals, as reported the 11th of May of 2020 [41]. This situation significantly threatened the sustainability of an already weakened health system, which struggled to provide health services to the increasing number of patients. Some have argued that the weakness of the health system can be attributed to its limited financial capacity, a problem that can be traced back to the economic crisis of 2008 when the financial support was decreased in response to the economic recession, and since then the national health system has remained underfunded [42]. To be able to meet the needs of the epidemic, health professionals decreased their regular activities at the hospital to focus all their working capacity on tackling the COVID-19 crisis, decreasing the capacity of non-urgent and specialized medical services.

Another concern during this situation was the limited detection capacity due to the lack of testing resources. At the beginning of April 2020, more than 45 days after the outbreak started in this country, there was still no rapid test available; PCR testing was the only method used to diagnose positive cases [43]. This method requires trained personnel, specialized equipment, and much longer time than the rapid tests, which were being used in other countries. As a result, testing was limited to severe cases needing hospitalization, and no testing was provided to contacts or symptomatic cases. Daily COVID-19 tests performed on 20 April 2020 were 0.23 per thousand people, this figure increased through the first wave of the pandemic to a maximum of 0.95 tests per thousand people during the month of May [44]. Similarly, testing was not provided to health workers that had been exposed to patients with COVID-19, resulting not only in dangerous conditions for workers themselves and the people in close contact with them, but to all other patients hospitalized for conditions not related to the current pandemic [45]. Another consequence of the lack of testing was the possible underreporting of cases, resulting in overestimated mortality rates being reported due to the lack of certainty on the real number of positive cases of COVID-19 in Spain. Finally, the lack of surveillance and case detection potentially caused further spread of the disease, as many of the unidentified cases were not taking the recommended isolation measures. Overall, the lack of testing resulted in a symptom-based strategy to control the disease, which was unlikely to succeed at stopping disease transmission due to the characteristics of SARS-CoV2, which has been reported to cause a high proportion of asymptomatic and mild cases.

In addition to the consequences seen in the healthcare sector, government interventions also had severe impacts on the lives of all the citizens. These impacts include the economic recession that the confinement measures will generate, causing an unprecedented situation which is predicted to have multiple short and long-term effects in the Spanish economy. Additionally, psychological consequences will also occur due to the restricted freedom, decreased social contact and persisting fear and insecurities caused by the health threat and the control measures; children, adolescents, and young adults are particularly vulnerable to these consequences given the important role of socialization at this stage in their lives [46]. Finally, this outbreak will also have a severe social impact, including an increased effect of the negative consequences experienced by vulnerable populations. Senior citizens, children and women at risk of violence, families and individuals at risk of poverty, migrants, socially excluded groups, people with low-paying or informal jobs are some of the groups that would be more severely affected by the psychological, economic, social and health consequences of the pandemic and the measures to control the spread of the outbreak, aggravating the existing inequalities across the population [47].

### 3.1. Economic Impact

Mobility restrictions and social distancing measures that have been implemented to contain the transmission of the virus have a direct and severe impact on the economic activity. The lockdown of the citizens resulted in an unprecedented situation, with all sectors of the economy experiencing a complete halt of production and consumption. These consequences have a strong impact on the activity of small and medium size enterprises, which in Spain account for about 61% of the economy [48]. To counteract this impact, the Spanish government established mechanisms to ease the economic impact of COVID-19 through six executive orders, which provided 200,000 million euros of support measures [33]. These measures included extraordinary credit for the health ministry; financial support, guaranteed supply of services and allowed payment delays for vulnerable families, workers and communities; protection of employment through regulatory methods which enabled the state to provide salaries for workers of affected businesses; and economic support to most severely affected sectors [49].

The exceptional economic consequences of COVID-19 crisis are estimated to have caused a contraction of the GDP of about 1.3% in 2020 according to Goldman Sachs [50]. The expected long-term consequences of this economic situation will result in a slowdown in the Spanish economy after the time of confinement due to an interruption of supply chains and the social impact caused by the outbreak, which will reduce domestic as well as external demand. The most severely affected sectors by this reduced demand are expected to be the tourism and transport sectors.

Moreover, the deficit generated during this period of lockdown will have an economic impact that can persist after the outbreak. Increased fiscal pressure on the citizens with measures such as increased taxes and decreased public spending are expected following the health crisis. The most economically vulnerable citizens, and those relying on small and medium enterprises particularly those on the hostelry, tourism and transport sectors will be the most damaged by the impact of the COVID-19 lockdown and the economic measures that will follow. Oxfam reported that the COVID-19 health crisis is expected to increase the unemployment rate from 13% to 19% of the population, and with the GDP contraction the rates of poverty could increase in 700,000 people, reaching 10.8 million people. This represents a 1.6% increase in poverty, from the 21.5% before the COVID-19 crisis to a 23.1% of the population living in poverty in Spain [51].

Luis de Guindos, vice-president of the European Central Bank, expressed his concern over the Spanish economy given its high economic dependence on the sectors most severely affected by the pandemic. On an interview during the first wave of the pandemic, he argued the economy could start recovering in 2021 although probably will not achieve full recovery during this year, and ultimately it will depend on the time the lockdown lasts in Spain. Given the impact of confinement on businesses, he emphasized the importance of providing temporary measures to help their sustainability. Public accounts will be extremely affected by this situation, adding a financial need of about 1 to 1.5 billion euros, “a figure never seen before”. To this public debt he claims it will be critical the intervention of the ECB. In face of this situation, he argues it would be essential to form a cooperative response from the euro zone for a more resilient and strengthened response to the economic recession [52].

### 3.2. Social Impact

One of the consequences which needs more consideration during this health crisis is the social impact on the population. Fear of infection and uncertainties about the duration and the consequences the pandemic will have on the citizens’ lives in the future, affect the wellbeing of the population. Appropriate communication through technology and social media allows information to be accessed by the population, making possible the delivery of reassuring messages to counteract the myths and misinformation, and ease the population facing this uncertain situation [53]. Another advantage of the abundance of information accessibility has been the awareness for shared social responsibility raised among the population, ensuring effective cooperation from all citizens, which made possible the implementation of the control measures established by authorities. For many citizens, a sense of collaboration also allowed the construction of a narrative of altruism around the confinement measures, which might have otherwise resulted on a perception of appropriation of freedom. To this positive environment of collaboration occurring during the first lockdown, it must also be mentioned the multiple initiatives that rose from individuals external to healthcare sector, contributing to the support of healthcare workers. From the simple initiative like communal applauses taking place every day at 8 pm, to the network of citizens that gathered their efforts to produce homemade equipment for protecting health workers delivering care to infected people. These initiatives not only provided many people with an opportunity to collaborate in the health crisis, but they also resulted in very effective contributions that ameliorated the work of health professionals and the situation of COVID-19 patients. Nevertheless, these actions of community cohesion, solidarity and cooperation among citizens started fading away as the measures were relaxed over the summer, and by the second wave of the pandemic the population started showing a less cooperative attitude owing to the burnout of the limitation of freedom, the fear of the spread of the virus and the consequences of the control measures, which have been taking place for longer time than most of us expected.

On a more negative note, there have been significant detrimental social effects resulting from this crisis, principally the increase in social inequalities. Thanks to the National Health System, which covers all citizens equally, healthcare access is not significantly affected by social inequalities. However, economic consequences of the lockdown and the closure of schools could have stronger impact on the most economically vulnerable. According to a report released by Acción Contra el Hambre, the health crises has caused an increase in the number of people employed by the informal sector and in low paying jobs with precarious conditions. Most people employed in these sectors are migrants for whom the opportunities of obtaining a formal job have been hampered by the health crisis and the slowdown in the process of regularization and residence or work permit renewal; as well as the reduced job opportunities in sectors commonly offering jobs for migrant populations such as the restoration and tourism sectors. In this report it is also noted how sanitary and personal protection measures are many times less strictly followed on low-paying jobs which can result in higher risk of infection from SARS-CoV2 among these more economically vulnerable populations [54]. The increased risks and challenges faced by more vulnerable groups already suffering social inequalities result in a growth of the gap of social inequalities accentuated by the consequences of the health crisis.

Young adults and children have particularly suffered the effects of the health crisis through the new educational policies and practices. It is estimated that about 6.7 million learners between 3 and 18 years of age were impacted by school closures in Spain at the end of the 2020 academic school year. At that moment, during the lockdown, educational activities were provided when possible through online platforms. However, the average percentage of schools capable of imparting online education was only 52% in the entire Spanish territory. There were significant differences in access to online learning across each community, varying from 71% in Castilla y León to 35% in Aragón. Moreover, these changes in educational activities unequally affected economically disadvantaged populations creating a digital gap, as under these circumstances a stable internet connection and access to technology resources is required in order to access the learning activities. In addition, how the schools could adjust the curriculum during the health crisis was an ongoing debate which caught schools unprepared at the end of the academic year, without a strategic plan on how to adjust to the situation. At that moment, there were also unresolved questions about how the next academic year would start in face of the sustained health threat [55]. The final plan for safely providing education services for the new school year starting on September 2021 was released the 27th of August, just few weeks before the academic year was planned to start [56].

Especially vulnerable to this health crisis is also the older population, and consequently there has been a severe impact in nursing homes for the elderly. These places are particularly vulnerable to disease outbreaks due to the high number of older people, which very often also present comorbidities and generally spend most of their time in a closed space and have close contact with caregivers and other residents. Since the beginning of the outbreak in Spain, some of these centers have been COVID-19 hotspots resulting in thousands of deaths that could have been avoided if a better preparedness and a response plan had been in place before the outbreak hit the country. In order to try to control the situation in these high-risk places, the government issued recommendations on the management of nursing homes, which include restriction of visits, isolation of cases and protection of caregivers with especial protective equipment [57].

Another significantly underestimated impact is the psychological consequences of the lockdown. According to an article that reviews the psychological effects of quarantine during previous health crises, there are multiple stressors that can considerably impact mental health of affected populations: mobility restrictions, lack of social contact, fear of infection, frustration, boredom, inadequate information, financial loss, and stigma. These can lead to psychological effects during and after quarantine, resulting in mental health consequences such as post-traumatic stress, confusion and anger. These effects can be minimized by reducing the time of quarantine, providing timely and adequate information and giving especial attention to the psychological needs of healthcare workers, which are especially vulnerable to the psychological effects. Psychological consequences need to be considered by decision makers in order to weight the impact of the interventions applied and try to maximize the wellbeing of the population [58].

Among the non-health impacts of the current pandemic, intra-family violence stands out as a social issue, which threatens to be aggravated as a result of the lockdown measures. Vulnerable people at risk of violence such as elderly, children and mainly women are forced to spend longer periods of time with their aggressors due to the confinement, and this can result in increased intra-family violence. Of special concern in Spain has been gender-based violence, data from the Government Delegation Against Gender Violence reported a 48% increase calls to the gender violence help line during the first weeks of April 2020 compared to the first weeks of March (before the lockdown); this represents a 47% increase when contrasted to the same period in the previous year [59]. In reaction to this situation, a contingency plan was developed to combat gender violence during the health crisis. This included a guide of action for women suffering gender-based violence and a campaign to inform and raise awareness on the services and resources available to women suffering gender-based violence [60].

Confinement measures also reduced physical activity levels among the population. According to data collected by FitBit, a 38% reduction on physical activity resulted from the lockdown measures implemented to control the outbreak during the first wave of the pandemic [61]. These measures impeded outdoor and exercise activities such as walking, jogging or biking; and can have detrimental effects on the health of the population, especially in children and people with comorbidities [62]. It is important to take into account these effects of the confinement, given the high prevalence and burden of disease caused by Non Communicable Diseases nowadays, and the important role daily exercise plays in the management of these diseases. In addition, it is well established that practicing daily exercise contributes to the wellbeing of the populations contributing to improve mental health in stressful situations. For children, who are in a period of development, being able to move, socialize and learn through games are necessary activities. Lack of these experiences can result in frustration, stress or anxiety, as children do not have the emotional and psychological resources to cope with confinement situation. According to a study published by Cambridge University Press, quarantine and isolation can create traumatic conditions for children and plans for pandemic control must particularly take into account populations with especial needs, particularly children, women, the elderly and people with disabilities [63].

On the other hand, an additional positive side effect to be noted is the improvement in air quality that resulted from the decreased traffic and industrial activity. Researchers claim this could contribute to protect vulnerable population against COVID-19 disease with higher risk due to respiratory comorbidities; as well as preventing deaths caused by air pollution. Another positive outcome of the measures established was the reduced number of traffic accidents that occurred during the time of restricted mobility [64].

## 4. Mathematical Model Predictions

Mathematical models developed by Universidad Carlos III in Madrid (UC3M), Universidad de Zaragoza, the Massachusetts Institute of Technology (MIT) and the Italian Foundation ISI analyzed the effectiveness of the measures implemented to control the spread of the COVID-19 in Spain as the first wave was taking place. The results concluded social distance measures are not effective if not taken in combination with proactive interventions such as massive testing, isolation of cases and contact tracing. The findings of this study state social distance measures and school closures are only effective when implemented for a long period of time, which result in increased undesirable negative effects as the previously mentioned economic, psychological and social impacts. It was also found that with exclusive implementation of passive measures, a new outbreak is likely to occur after these measures are released. Therefore, proactive measures such as active case finding, in combination with contact tracing and case isolation are necessary to avoid or minimize the impact of future outbreaks [65].

This study predicted the disease would remain among the population even if total confinement measures were implemented, which was the strategy applied in Spain at that moment. As it can be observed in Figure 1, this strategy results in a second peak of the epidemic after the restriction measures are lifted. Figure 2 and Figure 3 show the analysis of the probabilities of a subsequent epidemiological peak in the situation of six possible strategies: (S2) 90% social distancing, (S3) 70, 80 and 90% social distancing and school closure, (S4) restaurants and cultural activity closure, (S5) non-essential closure and (S6) total non-essential activity closure. Only strategy 3 with school closure and 90% social distancing implemented for 60 days has low probability of resulting in a successive peak. Alternatively, as depicted in Figure 3, after 90 days of implementation of the measures 80 or 90% social distancing and school closure can result in lower probability of a following peak, with 90% of social distancing showing an increased lower probability of a second peak (refer to the original paper for further details, methodology and explanations).

Nevertheless, it is important to take into account that these studies were developed using real-time data during the peak of the outbreak in Spain. Further studies with more complete data conducted later in time, can provide more specific and accurate picture on the different possible outcomes that could have resulted if different measures would have been implemented at the beginning of the pandemic.

## 5. Conclusions

### 5.1. Relaxation of the Measures

A question that was on the mind of Spanish citizens since the beginning of the lockdown measure–and it is still so after a year since the beginning of the pandemic— is when society will be able to go back to ‘normal’. This is a question that for now remains unanswered, and many agree that until 60 to 70% of the population is vaccinated or an effective treatment is developed, going back to normal will be difficult [66]. During the first wave, as the number of positive cases identified daily decreased, authorities started to lift the mobility restrictions in an escalated manner. This started the 26th of April when children under 14 years of age were allowed to walk for an hour daily accompanied by an adult and within one kilometer from their home or residence [67]. Although still restrictive, this considerably alleviated the negative impact many children were suffering from the confinement measures, which, as mentioned before, particularly affect these populations.

Lockdown relaxation and the transition to the new normality occurred in an asymmetric fashion in Spain, adapting to the evolution of the epidemic in each region with the aim to alleviate the social and economic impacts without overwhelming hospitals and ensuring they can operate within its capacity [68]. Authorities warned the lockdown release measures could be withdrawn when the number of positive cases increased and as predicted, after a short period of relaxation of lockdown measures, these started becoming more restrictive again through the last months of summer as the numbers increased, reaching the peak of the second wave the first week of November [69].

### 5.2. Discussion

In summary, the policies implemented in Spain to control the outbreak of COVID-19 at the beginning of the pandemic, focused on a rather passive strategy with strict total confinement measures as the main action taken to control the spread of the disease. This resulted in undesirable economic, psychological and social consequences, which could have severe and long-lasting effects on the population. Mathematical models predicted that in order to control further spread of the disease and prevent future outbreak peaks, a more proactive strategy needed to be implemented, including measures of active case finding, massive testing and contact tracing, as well as case isolation. Through the evolution of the pandemic the protocols have been adapted to introduce these measures in the Spanish strategy to control the spread of the pandemic. Nevertheless, as pointed out by the WHO expert deployed to Spain on a country support mission, increased resource availability and healthcare capacity and preparedness is needed in combination with restriction of activities to fight the spread of the virus [70]. This would require further scientific development and international cooperation for resource provision, as well as increased financial support of the healthcare system to allow an increased capacity to handle COVID-19 patients. A proactive strategy with interventions targeted to infected population could allow effective release of lockdown and restrictive measures and stop the aggravation of the negative impacts these measures are having on the population. It is important for the wellbeing of the population to reactivate the economic and educational activities, as safely and promptly as possible; to enable the population to go back to their regular social and professional activities, diminishing the impact this health crisis is having on their wellbeing. As the WHO expert concluded, the strategy implemented should find a balance between protecting health, reducing the socioeconomic impact, and defending human rights.

In conclusion, responding to a viral outbreak pandemic is a task which most policymakers were unprepared for, and the COVID-19 outbreak was an unexpected event whose effects were not predicted. Its impact has been more extensive than expected and its fast spread has forced countries to act ad lib, without the preparedness that would have been desirable in a situation of such broad impact on the population. Consequently, as cases increased exponentially in Spain, multiple measures of broad impact and unpredictable consequences were implemented in a seemingly improvised manner with the hope to minimize the number of positive cases and deaths. The struggles of different countries around the world to combat the viral outbreak have shown it is not easy to develop an effective strategy that will promptly and effectively control the outbreak, whilst minimizing the side effects on the population wellbeing. As it was stated by the WHO, “there is no single formula for which measures must be included or how to implement these, given the wide range of epidemiological situations and social and economic contexts in which the pandemic is occurring.” [71]. But there are surely better strategies which can minimize the impact of this health crisis on the population. On the ‘Covid-19 Strategy Update’ published by WHO, in addition specific recommendations of personal protection, and physical distancing measure to suppress community transmission; the need for a coordinated response based on a national plan was stressed, with especial attention to the coordination between national and subnational levels. This publication also reinforced the need to consider vulnerable sectors of the population when designing a strategy, to minimize the social impact of the pandemic. Victims of gender-based violence, children, senior citizens, migrants and workers of low paying jobs are among those that are at higher vulnerability of being negatively affected by the measures to control the Covid-19 pandemic [53]. Regarding the WHO recommendations, better coordination, preparedness and planning for a clear strategy, and increased focus on the social impact of the measures implemented could have enhanced the preliminary response to the outbreak in Spain.

In retrospect, given what is known now, Spain could have reconsidered its initial strategy and the measures taken, which might have prevented the high number of cases and deaths we have seen, or the harmful effects of the confinement on the wellbeing of the population. It is still possible to reconsider this strategy moving forward, and work to minimize the impacts this health crisis is having on the population. Moreover, reviewing the actions taken and its consequences is important in order to become better prepared to any threats or crises that could come in the future, particularly keeping in mind future waves of the outbreak. There is still time to learn from the current situation, and we must take the opportunity this pandemic provides to reinforce the national capacities and ensure improved preparedness in the future.

## Figures and Tables

**Figure 1 ijerph-18-04026-f001:**
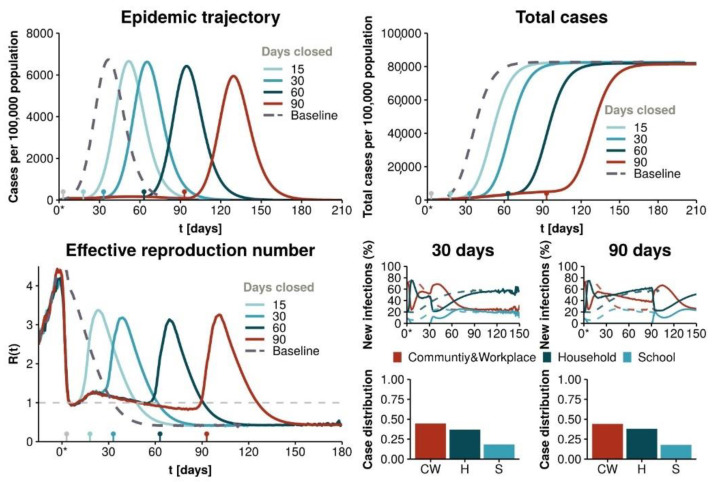
Mathematical model predictions for a strategy of total confinement [65]. Reproduced from [65].

**Figure 2 ijerph-18-04026-f002:**
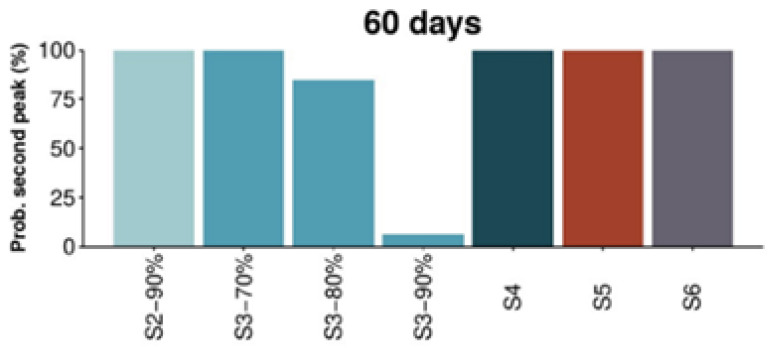
Probability of a second peak of the epidemic 60 days after measures are lifted [65]. Reproduced from [65].

**Figure 3 ijerph-18-04026-f003:**
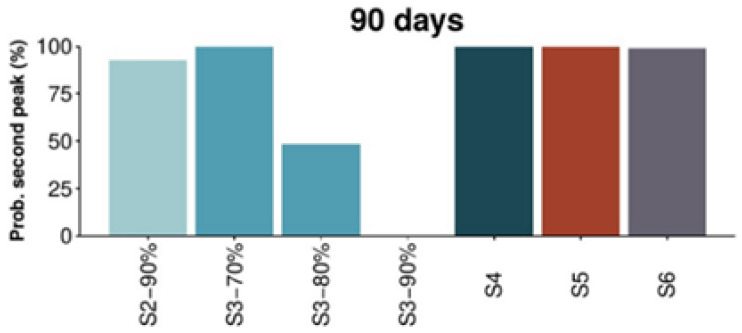
Probability of a second peak of the epidemic 90 days after measures are lifted [65]. Reproduced from [65].

## Data Availability

Data sharing not applicable.

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
