# Peer review of "Early Interventions and Impact of COVID-19 in Spain"

_ijerph, 2021, doi:10.3390/ijerph18084026_

Round 1

Reviewer 1 Report

The researchers present a case study on the effect of early intervention strategies for COVID-19 in Spain.  The mathematical predictions show the effects of containment as well as the probability of second peaks after measures are lifted.    The methods of modeling are sound, as well as the analysis and interpretation.   

Recommendations to authors:  

(1) Introduction - authors might consider citing published sources on the preliminary effectiveness of interventions in other parts of the world.

(2) The notation of certain graphs especially "S6" on Figure 1 should be removed, unless corresponding directly to figures or narrative elsewhere in the manuscript.        

(3) Wherever possible, the use of websites as source material should be replaced by citations from the peer-reviewed literature. 

Author Response

Thank you for the detailed feedback and helping improve this paper. I have made chages as suggested:

(1) Other case studies from the initial non pharmaceutical interventions in different countries (Brasil, Italy, India, Singapore, Switzerland, Greece, Iran, China, Egypt, Argentina, South Korea, and Australia) have been referenced in the introduction. See highlighted references.

(2) Good point. Footnote on the figure removed and added own footnote that corresponds with manuscript text.

(3) Agreed. This is a very good point and definitely one of the weaknesses of this report. However, this was written during last confinement when there was not much literature on the non-pharmaceutical interventions and their impact. And also, I decided to rely as much as possible on these first perceptions on the situation at the moment everything was initiating to capture as much as possible the impact at that moment and the perception of those, this is why unfortunately most references come from more "informal" sources rather than published papers that took place later on after the onset of  the pandemic.

Reviewer 2 Report

I recognize the efforts of the authors.
The paper has been improved by assertively responding to suggestions made.

Author Response

Thank you for the comments provided and helping this paper be more clear and comprehensive.

Reviewer 3 Report

The author/s have addressed my comments. the new version is much improved.

Author Response

(The authors gave the same response as above.)

Reviewer 4 Report

In section 3.2 Mathematical Model Predictions, 6 possible strategies were analyzed. However, the details of each strategy was not detailed. The authors should describe it. It is difficult to understand the meaning of “90% social distancing”. And the figure legend should be added in each figure.

And, “3.2 Mathematical Model Predictions” is better to be a separate section from "3 Management and outcome" and “3.2 Relaxation of the measures” is better to be included in "4 Conclusion and Discussions". Because the managements that had been taken and its outcome were described in other sections.  

Author Response

1. Description of the 6 strategies summarized in lines 443-445. Legends added (lines 451, 453, 455), and explicit reference to refer to the original paper for further clarification added - for the details and definitions of each strategy including 90% social distancing (line 449)

2. Good point. Thank you for the suggestion, mathematical model preditions has been formatted as a separate section and the relaxation of measures has been included within the conclusion section

This manuscript is a resubmission of an earlier submission. The following is a list of the peer review reports and author responses from that submission.

Round 1

Reviewer 1 Report

This article is written as a "case report" on early interventions and impacts of COVID-19 in Spain. Unfortunately, only secondary data from other sources are presented along with commentary.  In order to be ready to submit for publication, the manuscript would need to present some original material (e.g. researchers' own mathematical models) 

Reviewer 3 Report

This is an interesting and informative paper that provides a large amount of information on the reaction in Spain to the COVID-19 virus, and the consequewnces of social distance and preventive measures.

I think that the weakness is in the extent of information. It is difficult to follow what is different in Spain from other countries, that choose for strict measures. I was looking for the innovative angle or the interest "twist", something that happen in Spain and was different from other countries. For example the health and political system are de centralized, and will be interested to see the extent of consistency in policies and directions between the local and national goverment, between the goverments of different areas. It will be interesting to see a more comparative approach, how Spain differed from Italy, for example

How colaborative was the population of spain with socialpolicy? Any resistence to the social policy? by whom?

In other words my suggestion is to look for inconsistencies or controversies and at least provide the sense if concensus or conflict was dominant?

Reviewer 4 Report

Totally, this article is not scientific paper but a kid of reportage. The authors should show at least how to obtain the information source and analyze it.

This article includes 2 topics; one is the measurement for COVID-19 pandemic control implemented by government and its social and economic impacts, and another is Mathematical analysis. As the title of this article and the contents of abstract, the authors should focus on the intervention by government and its impact to society and economics in Spain. In mathematical model predictions, the formulas for each prediction shown in figure 1, 2 and 3 and the number that is the basis to develop each formula should be shown. So, 3.3. Mathematical Model Prediction section and related sentence in 4. Conclusions and Discussion should be deleted.

It is not clear what kind of measures and when the government implemented. So, table or figure, in which the timeline of government measures, the number of patients per day, and so on, should be added.

Line 62-64: the meaning of the article “60% of the … severe disease.” could not be caught. Please check the grammar.

Line 98-101: how about the acute care beds per total beds in Spain and in other European countries?

Line 117-118: after 25th of February of 2020, it better to add following sentence to the end of this sentence: ", one day after detection of first case".

Line 137-142: the authors should add the time line for economic measure implemented by government.

Line 180: in the limited testing capacity, it should be described some specific numbers, such as the number of inspections that can be accepted per day.

Line 189: is this “the” “it”?

Line 193-195: Based on the situation of Japan and United States, the limitation of test capacity might not affect very much. The test capacity of Japan was limited and that of the United States was better than Japan. However, the number of patients in the United States was increasing, though the number of patients in Japan did not increase as rapidly as in the United States. So, the authors should add some information in infectious disease control from the other aspect. A concrete number of how lack the testing capacity should be shown.

In section 3.1. Economic Impact, the description of economic measures is insufficient, so it is recommended to delete it or the economic measures in detail in relation to the situation of occurrence should be described.